# The Influence of Weather on the Occurrence of Aflatoxin B1 in Harvested Maize from Kenya and Tanzania

**DOI:** 10.3390/foods10020216

**Published:** 2021-01-21

**Authors:** Benigni A. Temba, Ross E. Darnell, Anne Gichangi, Deogratias Lwezaura, Philip G. Pardey, Jagger J. Harvey, James Karanja, Said M. S. Massomo, Noboru Ota, James M. Wainaina, Mary T. Fletcher, Darren J. Kriticos

**Affiliations:** 1College of Veterinary and Medical Sciences, Sokoine University of Agriculture, P.O. Box 3000, Morogoro, Tanzania; bentemba@sua.ac.tz; 2Queensland Alliance for Agriculture and Food Innovation (QAAFI), The University of Queensland, Health and Food Sciences Precinct, Coopers Plains, QLD 4108, Australia; mary.fletcher@uq.edu.au; 3Commonwealth Scientific and Industrial Research Organisation, GPO Box 2583, Brisbane, QLD 4001, Australia; 4Kenya Agricultural and Livestock Research Organization, P.O. Box 57811, Nairobi 00200, Kenya; wanjugu67@gmail.com (A.G.); jakakah79@gmail.com (J.K.); 5Tanzania Agricultural Research Institute, Arusha Road, P.O. Box 1571, Dodoma, Tanzania; lwezaura@hotmail.com; 6Department of Applied Economics, University of Minnesota, 1994 Buford Ave, 248 Ruttan Hall, Saint Paul, MN 55108, USA; ppardey@umn.edu; 7Department of Plant Pathology, Kansas State University, Manhattan, KS 66506, USA; jjharvey@ksu.edu; 8Department of Life Sciences, Faculty of Science, Technology and Environmental Studies, The Open University of Tanzania, P.O. Box 23409, Dar es Salaam, Tanzania; smsmassomo@yahoo.com; 9Commonwealth Scientific and Industrial Research Organisation, GPO Box 100, Canberra, ACT 2601, Australia; Noboru.Ota@csiro.au (N.O.); darren.kriticos@csiro.au (D.J.K.); 10Department of Microbiology, Ohio State University, Columbus, OH 43201, USA; wainaina.4@osu.edu

**Keywords:** aflatoxin, maize, risk, modelling, climate, Kenya, Tanzania

## Abstract

A study was conducted using maize samples collected from different agroecological zones of Kenya (*n* = 471) and Tanzania (*n* = 100) during the 2013 maize harvest season to estimate a relationship between aflatoxin B1 concentration and occurrence with weather conditions during the growing season. The toxins were analysed by the ultra-high-performance liquid chromatography-tandem mass spectrometry (UPLC-MS/MS) method. Aflatoxin B1 incidence ranged between 0–100% of samples in different regions with an average value of 29.4% and aflatoxin concentrations of up to 6075 µg/kg recorded in one sample. Several regression techniques were explored. Random forests achieved the highest overall accuracy of 80%, while the accuracy of a logistic regression model was 65%. Low rainfall occurring during the early stage of the maize plant maturing combined with high temperatures leading up to full maturity provide warning signs of aflatoxin contamination. Risk maps for the two countries for the 2013 season were generated using both random forests and logistic regression models.

## 1. Introduction

Contamination of foods and feeds by aflatoxin B1 and other mycotoxins continues to be of serious concern for human and animal health. Exposure of humans and animals to mycotoxins is mostly through the consumption of contaminated foods and feeds [1]. While acute cases of aflatoxicosis have been reported sporadically [2,3,4], the most common reported cases are of chronic exposure, where victims ingest sub-acute concentrations over a prolonged period [1]. Chronic exposure to mycotoxins is associated with a range of health conditions including cancer, immune suppression, reproductive disorders and nutritional and growth impairment, teratogenic and renal disease [5,6]. Kenya and Tanzania, like many other tropical developing countries, are highly affected by mycotoxin contamination, especially aflatoxins and fumonisins [7]. The toxins are a serious food safety concern with fatalities attributed to intoxication with the fungal toxins [8,9]. Mycotoxicosis in Kenya and Tanzania is associated with dietary intake, where maize and peanuts, both known to be highly susceptible to mycotoxin producing fungi, are significant components of local diets [10]. Approximately 132 million people across East Africa depend on maize as a staple food, and the extent of mycotoxin exposure has been correlated with daily maize consumption [11]. The magnitude of mycotoxin contamination in East Africa is thought to be exacerbated by low input agronomic practices, improper grain storage, extreme weather conditions and inadequate knowledge and action in control and management of the problem [12].

Kenya and Tanzania are characterised by varied topography, producing a complex variety of localized climatic conditions [13]. Both countries border the Indian Ocean and experience rising altitude as one moves away from the sea. In general, the areas near sea level experience warm, wet to dry weather, while the highlands experience sub-temperate cold climates. Between the two are varied weather zones of wet to dry and warm to cold climates. Environmental factors known to influence the occurrence of moulds and mycotoxins include the weather, especially precipitation and temperature [14] and other factors like soil types [15]. Weather plays an important role, together with other agronomic factors, in influencing fungal infection and mycotoxins formation in crops [14]. Temperature and rainfall are also important in ensuring crops are harvested in suitable dry or maturity status.

Maize consumption in East Africa is mostly through products prepared from maize flour, hence harvesting is ideally done when kernels are dry to below 14% moisture content [16]. Understanding the influence of weather variation is useful for designing appropriate control programs. This study was conducted to investigate how occurrences and amounts of aflatoxin B1 in maize at (or shortly after) harvesting from selected areas of Kenya and Tanzania relate to average temperatures and rainfall during the growing season.

Pitt and Miscamble [17] studied the germination and growth rates of isolates of *Aspergillus flavus* (the primary fungus responsible for aflatoxin production) in laboratory conditions controlling the temperature and water availability, ($a_{w}$), with a minimum $a_{w}$ requirement of 0.82 at 25 and 30 °C and 0.80 at 37 °C. Growth rates of fungal colonies were higher for the higher temperature regardless of the amount of water availability. Battilani et al. [18] developed a mechanistic model for the infection of maize by *A. flavus* to estimate the risk of aflatoxin contamination in the field. They related weather data—including temperature, relative humidity and rainfall—to field data collected on aflatoxin contamination in maize in Italy. Our study uses similarly available weather data to estimate aflatoxin production. We focused on a) the 100 days leading up to harvest, and specifically in the period 100 to 34 days before harvest, which aligns with kernel development to maturity, and b) the final 34 days before harvest, during which the cob matures in the field.

## 2. Material and Methods

### 2.1. Study Area and Participants

In this study, dry maize grain samples were collected from pre-identified small-scale farmers’ stands and households in different parts of Kenya and Tanzania in the 2013 harvest seasons. Samples were collected during harvest or a few weeks thereafter. Farms were chosen that were closest to randomly selected locations in each county/region, with a frequency weighted by the maize production for each county or region. The survey intended to collect 600 samples from each country, but logistics and other factors reduced the final usable sample size to 571 in total, all with requisite farm information and measured aflatoxin values. District agricultural officers were trained as enumerators. Participants were briefly educated on the objectives and the importance of the research prior to planting seasons then asked to voluntarily consent to take part in the study. Ethics approval for the survey was granted by the Commonwealth Scientific and Industrial Research Organisation (CSIRO) Social and Interdisciplinary Science Human Research Ethics Committee (ID: 009/12).

### 2.2. Collection of Maize Samples

In total, 100 samples were collected from farmers in 12 regions in Tanzania and 471 samples were collected from farmers in 38 counties in Kenya. On average, about 0.5 kg of maize kernels were collected from at least five cobs of maize taken from different points within the maize stand. Dry maize kernel samples were put in paper bags that were sealed by stapling and then wrapped with rubber bands and transported to regional research institutes for pre-processing prior to shipment to a central laboratory for standardized aflatoxin analysis. The decision to collect kernel samples from the field contributed to a higher cost of sample collection but avoided the possible bias associated with storage contamination when samples are collected from more convenient locations such as stores or posho mills [19].

### 2.3. Preparation of Maize Samples

Preparation of the samples included milling and sub-portioning. Before milling each kernel, the sample was spread on a clean tray and large contaminant particles such as stones and pebbles were removed by hand. Dusty samples where further cleaned by sieving. Samples were then milled into flour using Romer Series II^®^ mill. The mill was set to perform automatic sub-portioning of the output flour into three sub-samples collected in clean, new zip lock plastic bags. After milling each sample, thorough cleaning of the milling chamber was done by vacuuming for approximately 30 s, with an additional ca. 50 g (one handful) of the next sample milled and discarded to avoid cross-contamination. The sub-portions of each milled maize samples in zip lock plastic bags were placed together in one paper bag and stored at 4 °C ready for aflatoxin B1 analysis.

### 2.4. Chemicals and Reagents

High-performance liquid chromatography (HPLC) grade methanol, acetonitrile, formic acid and ammonium formate (supplied by Sigma-Aldrich^®^, St. Louis, MO, USA). HPLC grade Milli Q^®^ water was supplied by Biosciences East Central Africa (BecA), International Livestock Research Institute (ILRI), (P.O. Box 30709 Nairobi 00100, Kenya). Aflatoxin B1 was analysed by a multiple mycotoxin analysis method with mixed mycotoxins standards (Aflatoxin B1, B2, G1 and G2, Fumonisin B1 and B2, T-2 toxin, HT-2 toxin, and diacetoxyscirpenole) obtained from Sigma-Aldrich (Sigma-Aldrich Chemie B.V., Zwijndrecht, The Netherlands).

### 2.5. Mycotoxin Extraction

Aflatoxin B1 was determined by simple solvent extraction and ultra performance liquid chromatography, mass spectrophotometry (UPLC-MS/MS) analysis in an adaptation of a previously described method [20]. Milled maize (5 g) was extracted with an extraction solvent (20 mL) comprising acetonitrile/Milli Q^®^ water/formic acid (790/200/10). The mixture was shaken mechanically using a mechanical orbital shaker (New Brunswick Scientific, Edison, NJ, USA) for approximately 90 min. The tubes were then centrifuged at 3000 rcf for 2 min. A 0.5 mL aliquot of the supernatant was transferred into a 1.5 mL Eppendorf^®^ tube and mixed with 0.5 mL mobile phase A (water/formic acid; 99/1 in 10 mM ammonium formate). The mixture was gently shaken and filtered through a 0.2 micron polyvinylidene fluoride (PVDF) syringe filter into 2 mL glass vials for analysis by UPLC-MS/MS. Where analysis was not done immediately, the extracts (in vials) were stored at 4 °C and analysed within 24 h.

### 2.6. Aflatoxin Detection and Quantification by LC-MS/MS Method

The UPLC-MS/MS system consisted of Shimadzu ultra-high-performance liquid chromatography (Shimadzu^®^ UHPLC Nexera, Shimadzu, Nishinokyo Kuwabara-cho, Nakagyo-ku, Kyoto 604-8511, Japan), coupled to ultra-high sensitivity, ultra-fast triple quadrupole, tandem-mass spectrophotometer (Shimadzu^®^ 805, Shimadzu), equipped with an electrospray interphase (ESI). Chromatography was performed on a Synergi^®^ Hydro RP (100 × 3 mm, 2.5 µm) column (Phenomenex^®^, 411 Madrid Avenue, Torrance, CA, USA) at 40 °C column temperature with an elution gradient composed of mobile phase A (Water/formic acid; 99/1 in 10 mM ammonium formate) and mobile phase B (Methanol/water/formic acid; 97/2/1 in 10 mM ammonium formate) at a flow rate of 0.5 mL min^−1^ and a gradient as follows: 0–2 min, 98% A–60% A; 2–4 min, 60% A; 4–6 min, 60% A–40% A; 6–10 min, 40% A; 10–12 min, 40% A to 0% A; 12–16 0% A. The mass spectrophotometer was operated in ESI positive mode using the following conditions: heating block temperature 400 °C, interface temperature 300 °C, desolvation line temperature 250 °C, interface voltage 4.0 kV, conversion dynode 10.0 kV, nebulizer gas (nitrogen) flow 180 Lh^−1^, collision gas (argon) flow 600 Lh^-1^, heating gas (air) flow 600 Lh^−1^ and collision-induced dissociation (CID) gas (argon) pressure 270 kPa. Aflatoxin B1 was quantitated by using multiple reaction monitoring (MRM) transitions of 312.80 → 285.15 and 312.80 → 241.15. The LC-MS/MS quantitative analysis was conducted by LabSolutions LCMS Ver 5.6 computer program which was used to generate raw data for aflatoxin concentrations.

### 2.7. Weather Data

The weather data were downloaded from the Global Surface Summary of the Day (GSOD) data provided by the US National Centers for Environmental Information (NCEI) using the R package GSODR [21]. The database provides daily summary information for precipitation, temperature and relative humidity among others collected from 31 weather stations located in Kenya and Tanzania and one in neighbouring Uganda. The maize samples tested for aflatoxin were grown during the 2013 growing season, planted sometime during the months of March, April or May. Weather information was extracted from stations for the 100 days before harvest, i.e., from the 24 April 2013 until the 31 July, both inclusive. This aligns approximately with maize cob development and maturity. Estimates of daily precipitation, mean temperature and relative humidity for each site were calculated by spatio-temporal kriging using the krigST function in the R package gstat [22,23]. The average daily temperatures, average daily rainfall and average daily relative humidity for each farm used in the statistical models were calculated for two periods of the maize maturing stage, the first representing early and the second, the late maturing stage.

### 2.8. Data Analysis

The distribution of aflatoxin levels in the maize samples was highly right skewed, with a prevalence rate of only 29%. We chose to consider this as a classification problem, i.e., the prediction of aflatoxin occurrence in samples, based on weather variables. We tested a suite of modelling approaches to explore the performance of these methods targeting their predictive capabilities based on the estimated weather features. These methods included support vector machines (SVM), random forests (RF), classification trees (CART) and linear discriminant analyses (LDA). We also fitted the more commonly used logistic regression model as a parametric approach. This allowed us to gain insights into the possible relationships between weather drivers and the occurrence of aflatoxin in maize. The predictive measures compare the predicted ability of the different modelling approaches.

The covariate design space was complex, so the weather measurements were categorised as factors of 0.5 mm width for rain, 2 °C for temperature and 2.5% for relative humidity. The starting model included main effects as second order polynomials for each of the weather classes. We then used a backward step variable selection method [24] employing the Akaike Information Criterion (AIC) statistic, the F and $\chi^{2}$ statistics where appropriate to find the parsimonious models that accurately estimated the aflatoxin level and incidence in the samples from the weather features.

The main measures we used to compare the models were accuracy, sensitivity, specificity, positive predictive value and negative predictive value. Positive predictive value is the probability that a sample showing a positive test result contains aflatoxin and the negative predictive value is the probability that an aflatoxin-free sample shows a negative test result.

Using the chosen statistical models, we also estimated the risk of aflatoxin occurrence in the 2013 maize crop using gridded weather data across all areas of Kenya and Tanzania. The estimates of weather across this grid were produced using the same kriging models used to estimate daily weather estimates for the sample points. Since the model is based on easily measurable and possibly forecast values, our aim was to find a model that can estimate aflatoxin occurrence in maize experiencing similar weather. In Kenyan maize growing areas, the dry mid-altitude country east of Katumani represents a higher risk area, while in Tanzania, the coastal regions and north western regions, the tropical savannah climate areas of Tanzania represent high risk areas [25]. Statistical analyses were performed using the R [26] program.

## 3. Results

### 3.1. Incidence and Concentration of Aflatoxin B1

Aflatoxin B1 was analysed in the 571 collected maize samples by the previously validated LC-MS/MS method [20], with aflatoxin standards across six concentration levels 0.2–21.1 µg/L demonstrating good linearity R^2^ > 0.996 throughout the analysis and a limit of quantitation (LOQ) of 0.6 µg/kg. Recovery efficiency and reproducibility of this analytical method were demonstrated by replicate analyses (*n* = 6) of the matrix spiked at six concentration levels (5–200 µg/kg), with recoveries of 77–105% and relative standard deviations below 29%.

Table 1 gives the level of aflatoxin B1 (µg/kg or ppb) and observed occurrence or incidence (%) of affected samples in the survey (limit of detection 0.2 µg/kg). In about half of the 38 sampled counties in Kenya, the average aflatoxin B1 concentration was below 1 µg/kg. Incidence represents the percentage of samples collected in a region in which aflatoxin concentration were in excess of 0.2 µg/kg. Thirty percent of the samples collected were contaminated with aflatoxin B1.

Observed proportion of samples with detectable levels of aflatoxin from different counties of Kenya and regions of Tanzania are shown in Figure 1. Incidences for regions from which more than 4 samples were collected are shown.

The statistical distribution of aflatoxin concentration was extremely right skewed (see Figure 2). Seventy percent of samples were aflatoxin free (less than the limit of detection, 0.2 µg/kg) with a small number of high values, while 88% of all the samples spanning both countries had less than 10 µg/kg, with the maximum being higher than 6000 µg/kg. Statistical inference based on linear regression models rely on the distribution of samples being reasonably symmetric, and with 79% of all values less than 1 µg/kg, the log transformation was not sufficient to normalise the distribution of the highly skewed response. Reducing the aflatoxin measurement to a binary response (presence/absence or occurrence) and fitting a logistic regression model is an option that avoids some of the distributional properties needed to fit linear regression models.

### 3.2. Weather Observations during Crop Growth

Most of the reported sowing dates for the maize crops from which the samples were taken ranged from March to May, while the harvest dates mostly occurred in July and August. Neither the exact dates of harvest or sowing nor the specific identity of the varieties sown were available for the sites. We initially focused on two periods of weather observations, a 67-day period representing the vegetative growth stage, and a 33-day period approximating the grain filling stage.

Average daily rainfall, humidity and temperature were recorded at 31 weather stations across Kenya and Tanzania for more than 100 days during the growing period of May 2013 through to July 2013. The maximum average daily precipitation (rainfall) was 3.21 mm. The average daily temperature ranged between 16.2 to 25.7 °C. The average relative humidity ranged between 51.4% and 89.3%. The median distance between survey locations and the closest weather site was 66 km with the closest weather station being as close as 3 km away from one of the surveyed locations.

We fitted separable exponential spatio-temporal models to explain the covariance existing between daily weather measurements from the weather stations. Kriged estimates of the 100 daily relative humidity, temperature and rainfall values at the 571 sampling locations were generated as described above.

### 3.3. Statistical Models for Incidence of Aflatoxin

The average values of daily rainfall, humidity and temperature for the two time periods leading up to harvest for each location were calculated and used as predictors in the various classification models. The R package caret [27] uses the train function to evaluate, using resampling, the effect of model tuning parameters on performance choice, chooses the “optimal” model across these parameters, and calculates model performance using a test sample. We chose overall classification accuracy (for the presence of aflatoxin) as the measure of performance. The modelling methods trialled included logistic regression (LR), classification and three machine learning (ML) methods, namely regression trees (CART), support vector machines (SVM) and random forests (RF).

The performance of the tested models is given in Table 2.

The logistic regression modelling approach required judicious simplification of the model for assessing the incidence of aflatoxin in samples. Average daily temperature, average relative humidity, average daily rainfall and average maximum daily temperature for the two growing periods were categorised as ordered factors. A model involving linear trend terms representing average daily rainfall for the early plant growth stage period and average daily temperature for the period prior to harvest was constructed. The logistic regression model provides some understanding of the relationships between the climate drivers and aflatoxin presence, while the ML methods focus on maximising predictive skill.

The random forest model achieved an overall accuracy of 80%, with a 95% confidence interval (CI) of (75%, 84%), while the accuracy for the logistic regression model is 65%, with a 95% CI of (60%, 70%). The random forest model achieved higher levels of specificity and sensitivity compared with the logistic regression model outperforming the other models in terms of its ability to accurately estimate which samples were true positive and which were true negative. Approximately 77% of the samples estimated to be positive for aflatoxin contained aflatoxin for the RF model and 68% for the LR model, while the logistic regression model declared approximately 37% of the contaminated samples to be free of aflatoxin, compared with 18% for the random forest model.

As shown in Figure 3 aflatoxin was observed in maize samples more frequently at sites with lower rainfall in the early stage of maturity, and higher temperatures for the later stage. The estimated parameters for the logistic regression model agree with these observations. Table 3 provides the odds ratios and confidence intervals for the final logistic regression based on the upscaled sample. For each increase of 0.5 mm in daily rainfall in the early maturing stage, the likelihood of aflatoxin almost halves (odds ratio = 0.56), while for every increase of 2 °C on average daily temperature in the later maturing stage this likelihood increases by 57% (Odds ratio = 1.57). The variable importance statistics from the random forest model are listed in Table 4. Average daily rain during the early maturing phase was considered important in both the logistic regression and random forest models.

### 3.4. Estimated Probability of Aflatoxin in the 2013 Season

We fit random forest and logistic regression models to estimate the likelihood of occurrence or risk of aflatoxin in maize for all of Kenya and Tanzania for the 2013 growing season. We used the same weather observations from the GSOR website and the same spatio-temporal models to spatially krige a grid of points across the extent of Kenya and Tanzania (*n* = 638) for each day of the 100 days leading up to 1st August 2013.

The likelihood of aflatoxin in maize is plotted in Figure 4. The differences in the observed (Figure 1) and estimated occurrences (Figure 4) are to be expected given the low percentage of samples with aflatoxin, the sampling variation associated with aflatoxin levels in maize, and the use of kriged weather information from weather sites located some distance from the sampling sites.

## 4. Discussion

This study recorded marked variation in the incidence and concentration of aflatoxin B1 in maize samples collected from quite dispersed sites throughout maize growing areas in Kenya and Tanzania in 2013. Both classification methods show higher risk areas in Kenya and Tanzania to be the arid steep hot region east of Machakos in Kenya and the tropical savannah regions of Tanzania [25]. The two methods estimate different risks for the dry winter, warm summer temperature region, and the Serengeti Plain area west of Arusha.

Several previous survey studies conducted in the two countries recorded high prevalence of the toxin in maize and related products [28,29,30,31]. Incidence and concentration of aflatoxins in maize in the East Africa region have been a central focus of research on mycotoxin contaminations of foods due to the potential and observed health impacts of the toxin and the nutritional importance of maize. Although country survey reports for mycotoxin occurrence in Kenya and Tanzania could not be located, following the aflatoxicosis outbreak in eastern Kenya in 2004, a cross-sectional survey study was conducted around the affected areas and reported that 55% of the maize obtained from markets was contaminated with aflatoxins above 20 µg/kg [32]. In Tanzania, health effects associated with aflatoxin exposure have been a topic of concern since at least the 1970s [33], and in 2008 the occurrence of aflatoxin in maize was reported in a study that also reported the presence of fumonisins in the samples [34]. The majority of recent studies indicate incidences of aflatoxins contamination to be above 25%, with a significant proportion of the samples having concentrations above acceptable levels [10,30,31,34]. In this study, about 30% of the maize samples were shown to have detectable levels of aflatoxins, 12% of which were above 10 µg/kg.

Understanding general contamination load in terms of incidence and concentration of mycotoxins is a first step in dealing with the problem. However, further epidemiological studies to characterize the occurrence patterns are important because mycotoxin contamination is influenced by multiple factors [35]. This study reveals substantial variation in the occurrence load of aflatoxin B1 between regions. This is more obvious with the samples from Kenya, where aflatoxin incidences and amounts are generally higher in the eastern parts of the country compared with the central and western parts. In Tanzania, the surveyed regions in the eastern part of the country also have samples with high average level of aflatoxin B1. The regional pattern of variation we reveal in this study is not as clearly evident in previous survey studies as most of them involved sampling frames that were more spatially constrained in different parts of the country. However, one study conducted during the acute aflatoxicosis outbreak in 2004 found that both the incidence and amount of aflatoxins in maize collected from markets in Makueni were higher than other markets located in the Thika area [32]. Makueni is located in the eastern part of the country and is among the areas affected by acute aflatoxicosis, while Thika is located in the central highland regions of the country. The findings by Yard et al. [36], suggesting that people in the Nyanza and Rift Valley regions were less exposed to aflatoxins (by testing blood indicators), align with the suggestion that despite high levels of contamination observed in peanuts [31,37], the community in western Kenya might be at lower risk of consuming contaminated grain as compared with those located in the Eastern and Coast regions. A study of maize collected from 243 posho mills in eastern and western Kenya in 2009 and 2010 reported that 60% of the 2466 samples had detectable levels of aflatoxin and 28% of these samples had levels above the regulatory limit of 10 µg/kg [19].

Several prior studies have demonstrated the relationship between aflatoxin occurrence and environmental conditions by indicating that a short-term increase in temperature or reduced soil moisture may lead to increased levels of contamination [38], which in turn might be associated with fungal responses to increased expression of the aflatoxin synthesis genes [39]. The effects of short-term and long-term variation in weather conditions on aflatoxin production are not clearly established in the prior literature. Smith et al. [19] provide evidence of some association between presence of aflatoxin in samples collected from eastern and western Kenyan posho mills with normalized difference vegetation index (NDVI) and soil characteristics including soil organic content and cation exchange capacity. They did not find any association with rainfall (pre-flowering or post-flowering) however any such effect may have been suppressed by the presence of NDVI in the model. Interestingly, the association between NDVI and the presence of aflatoxin in this study were in different directions between pre-flowering and post-flowering NDVI measurements.

The regional variation observed and the weather associations established in this study are in accordance with the notion that high incidences and amounts of aflatoxin occurrences are associated with low annual rainfall and high average temperatures. For each millimetre increase in daily rainfall in the early maturing growth stage of maize, the likelihood of aflatoxin drops four-fold, while for every degree increase on average daily temperature in the later maturing stage the likelihood of detectable aflatoxin contamination increases by a factor of a little less than 30%.

## 5. Conclusions

This study provides some evidence of the relationship between the occurrence of aflatoxins and weather conditions for one season of maize crops in East Africa. Low rainfall occurring during the early stage of the maize plant maturing combined with high temperatures leading up to full maturity provide warning signs of aflatoxin contamination in maize growing in this region. Our results provide direction for further investigation into the use of statistical models to forecast the production of aflatoxin by *Aspergillus flavus* in maize crops using the weather features investigated in this study.

## Figures and Tables

**Figure 1 foods-10-00216-f001:**
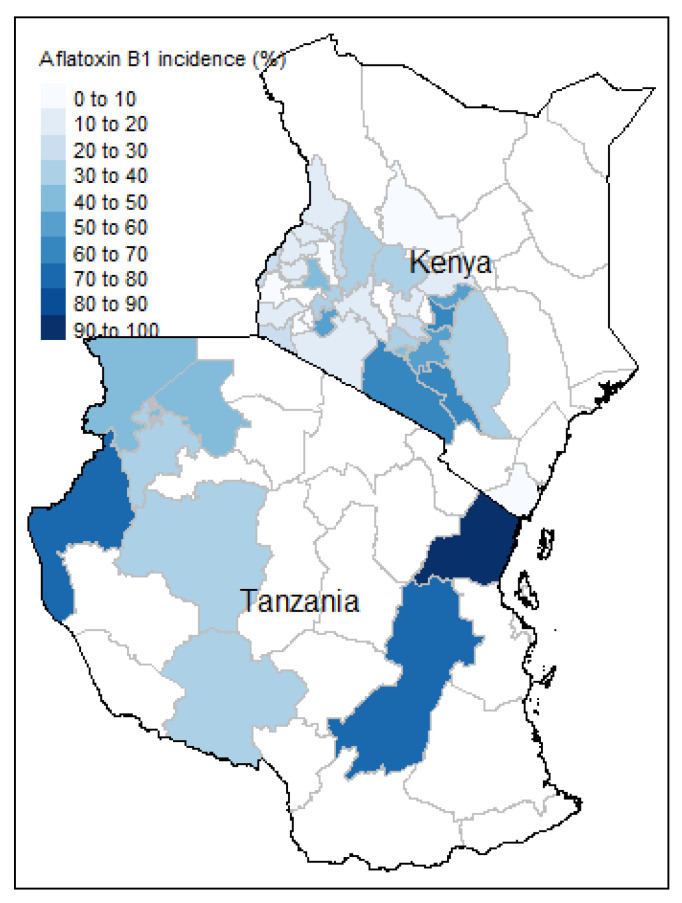
Proportion of samples positive for aflatoxin for the 41 of the regions/counties surveyed that had at least five samples.

**Figure 2 foods-10-00216-f002:**
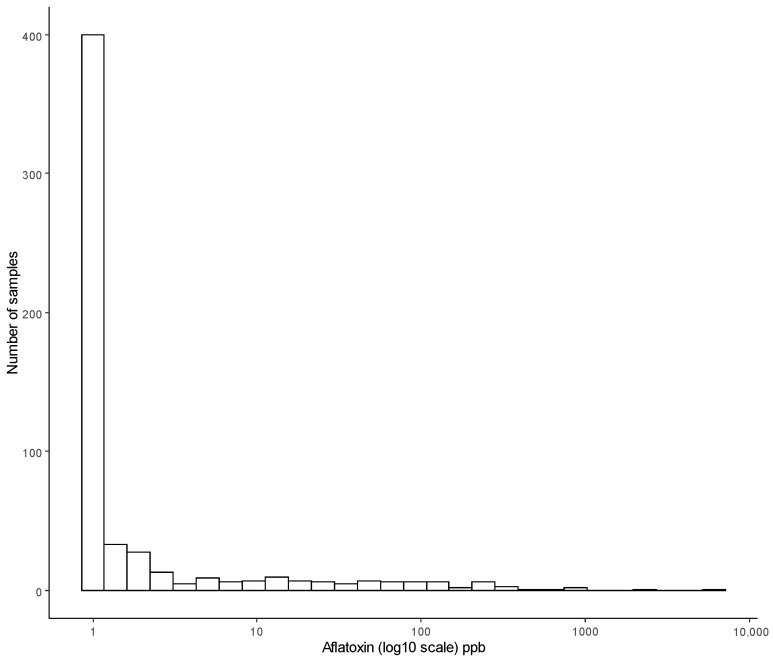
A histogram showing the distribution of aflatoxin values measure on 571 maize samples taken from farms in Kenya and Tanzania.

**Figure 3 foods-10-00216-f003:**
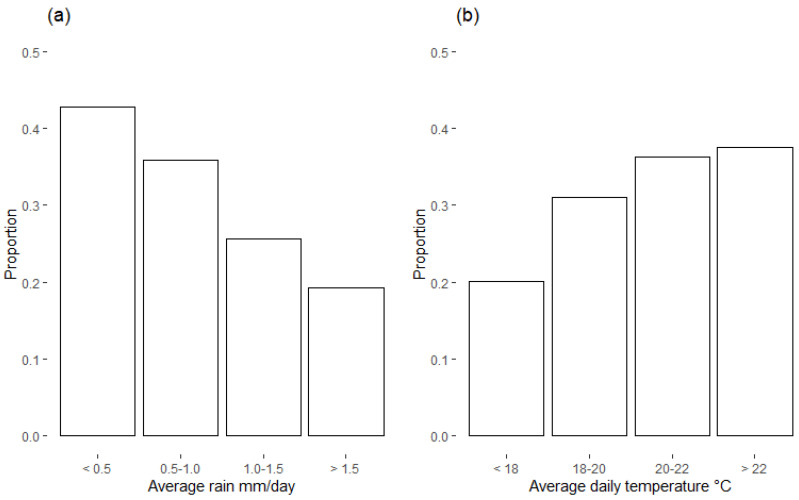
Proportion of samples with detectable aflatoxin (above limit of detection 0.2 µg/kg) for various levels of early rain (**a**) and late temperature (**b**). ‘Early’ refers to the first 67 days of the growing season and ‘late’ refers to the last 33 days of the maize growing season. Lower rainfall (early stage) tended to be associated with a higher proportion of aflatoxin-contaminated maize samples, while higher average daily temperatures (later stage) were associated with an increased proportion of contaminated samples.

**Figure 4 foods-10-00216-f004:**
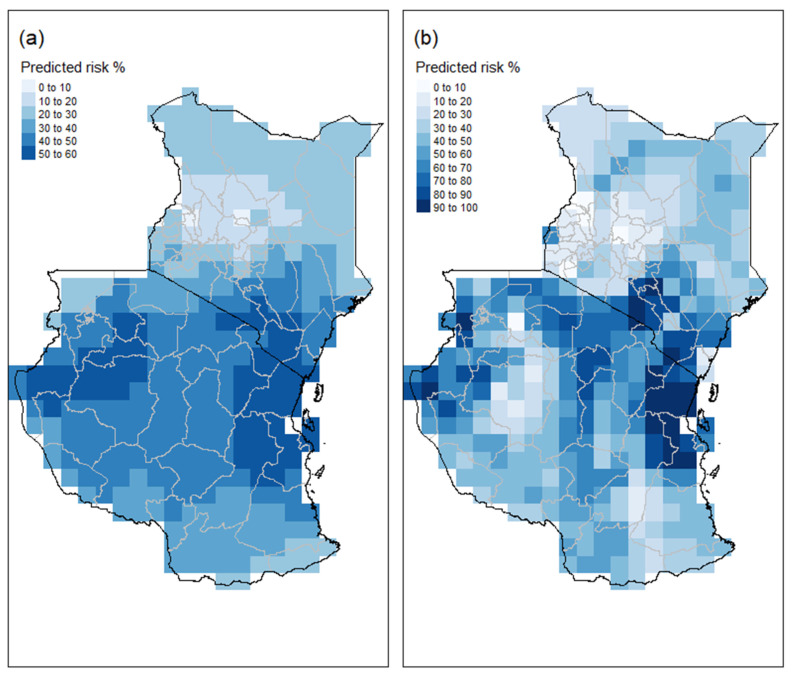
Estimated aflatoxin risk in maize for East Africa for the 2013 season. (**a**,**b**) Likelihood of harvested maize samples containing aflatoxin based on the logistic regression and random forest models, respectively.

**Table 1 foods-10-00216-t001:** Summary statistics for aflatoxin measurements for the entire survey. “Incidence” is the percentage of samples that had detectable levels of aflatoxins, whereas “level” refers to the concentration of aflatoxin among all samples with detectable levels of aflatoxin (level of detection, LOD 0.2 µg/kg).

Number of Samples	571
Incidence	29.4%
Median level of aflatoxin of samples with detectable level of aflatoxin i.e., greater than LOD (0.2)	4.775 µg/kg
Maximum level of aflatoxin	6074 µg/kg
Number of counties surveyed in Kenya	38
Number of regions surveyed in Tanzania	12

**Table 2 foods-10-00216-t002:** Measures of classification performance for a test set using logistic regression and machine learning models built from an upscaled data set.

Measures	LR	CART	SVM	RF
Average Accuracy %	64.9	64.1	70.2	80.0
Sensitivity %	57.5	56.4	60.8	82.9
Specificity %	72.4	71.8	80.0	76.2
Pos Pred Value %	67.5	66.7	74.8	77.2
Neg Pred Value %	63.0	62.2	67.0	81.6

**Table 3 foods-10-00216-t003:** Odds ratios and their 95% confidence intervals for parameters from the logistic regression model for the presence of aflatoxin.

Parameters	Odds Ratio	2.5%	97.5%
Intercept	1.579	0.819	3.062
Early rain	0.556	0.441	0.695
Late temp	1.574	1.298	1.920

**Table 4 foods-10-00216-t004:** The relative importance of variables in the random forest mode (RH, relative humidity).

Variable	Relative Importance
Early rain	100.0
Early RH	77.9
Late rain	29.6
Late temp	26.8
Early temp	8.5
Late RH	0.0

## Data Availability

Data included in these analyses required exact locations to be used to harmonize aflatoxin and weather information. As part of the ethics approval for this study, we are unable to provide these data that may identify surveyed farmers.

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
