# Peer review of "The Influence of Weather on the Occurrence of Aflatoxin B1 in Harvested Maize from Kenya and Tanzania"

_foods, 2021, doi:10.3390/foods10020216_

Round 1
Reviewer 1 Report
Summary
The authors present the results of a modelling exercise to relate weather station data to aflatoxin levels in maize collected from farmers in Kenya and Tanzania. This paper presents both the collection of the samples as primary response data, and collection and analysis of weather data as the independent variables. Various machine learning models are compared to logistic regression.
General Comments
Overall, this study generates useful data to explore the impact of weather during the late growing season and near harvest on aflatoxin contamination in Kenyan and Tanzanian maize. The underlying data collection and statistical modelling appear valid and useful. But there are some critical omissions and weaknesses in the writing that must be addressed.
- Three tables are missing, ID’d as ??. Line 171 (aflatoxin levels by sample), Line 231 (odds ratios), and line 236 (variable importance). Each of these tables are critical to the study. If their lack was simply a technical error (on either the author or reviewer side), I hope the editor can note this and facilitate a rapid re-review.
- The 5 fitted models could be presented more clearly. As written, the reviewers sees Table 1 as comparing mean accuracy of the 4 machine learning models. Then Table 2 compares Sn, Sp, PPV, and NPV for Logistic regression and Random Forests. The logic for this presentation is unclear. My impression is the authors first chose the best ML model, random forests, based on accuracy. Then chose to compare that best model to logistic regression. If so, that should be clearly written. Regardless, I would suggest making a single table with all 5 models (4 ML model, and LR), and all 5 metrics (Acc, Sn, Sp, PPV, NPV).
- As written, it was not clear if the authors controlled for other covariates that could influence mycotoxin contamination in difference regions. For example, different regions of each country could have different customary growing practices, typical maize lines, levels of income, supply chains, soil types, etc. Without comment on such other possible causes of differences, this work is about correlation, not causation. While the authors are carful to only talk about association, it would be good to comment if it is appropriate to base a forecast model (last conclusion sentence) purely on association.
Minor Comment
52-62: This introduction presents climate and weather work in Europe but does not mention any other work on weather and mycotoxin in Africa. Nor was any other work mentioned in the discussion. The reviewer would suggest searching for additional relevant literature. One paper comes to mind: Smith, L., M. Stasiewicz, R. Hestrin, L. Morales, S. Mutiga, and R. Nelson. 2016. Examining environmental drivers of spatial variability in aflatoxin accumulation in Kenyan maize: Potential utility in risk prediction models. African J. Food Ag. Nut. Dev. 16.. http://doi.org/10.18697/ajfand.75.ILRI09
Reviewer 2 Report
The findings shown in this paper are interesting. No objections at all regarding the regression techniques that have been studied, the results and the derived discussion. The main concern of this reviewer deals with some technical aspects related with the aflatoxin analysis and the results obtained:
Part 2.2.- Can you provide some further details about any sampling plan adopted? Samples were taken just at random? Any criteria about cobs or kernels visually damaged or affected by fungi in the field?
Analytical procedure for aflatoxin B1.- Please. can you provide any further information about the validation of the analytical procedure? The only data is in the text of Table 1 where is said that the average recovery was 79 %. What level or levels were checked for recovery? What about precision? What about measurement uncertainty? The LOQ was really checked in practice?
Part 3.1.- What was the value given for aflatoxin B1 concentration when the toxin was detected above the LOD and below the LOQ? What was the real value for the incidence? In Table 1 is 30.65 % and 30.2 % in the Abstract. Would it be possible to prepare a bit more detailed set of results? In line 325 is said that 12% of samples were contaminated above 10ppb. What else?
The following remarks are mainly editorial:
1.- The names of fungi are usually written in italics (i.e. line 78)
2.- Is rightly written the sentence starting in line 81?
3.- Line 154: The reference should be placed at the end of the sentence.
4.- Table 2: Mean accuracy for RF was 77.4 %. In line 260 is said that RF model achieved an overall accuracy of 75 % (same in the Abstract).
5.- References.- Authors must review in depth this Section: Capital letters are lacking in a lot of country names, authors and Organizations. Italics are lacking in names of fungi, etc.
Round 2
Reviewer 1 Report
Good work on the revisions, they fully addressed my comments
Author Response
Reply to both reviewers comments in attached document
